# Efficient and Effective Optimal Transport-Based Biclustering

**Chakib Fettal**
Centre Borelli UMR 9010
Université Paris Cité
Informatique Caisse des Dépôts et Consignations
chakib.fettal@etu.u-paris.fr

**Lazhar Labiod**
Centre Borelli UMR 9010
Université Paris Cité
lazhar.labiod@u-paris.fr

**Mohamed Nadif**
Centre Borelli UMR 9010
Université Paris Cité
mohamed.nadif@u-paris.fr

## Abstract

Bipartite graphs can be used to model a wide variety of dyadic information such as user-rating, document-term, and gene-disorder pairs. Biclustering is an extension of clustering to the underlying bipartite graph induced from this kind of data. In this paper, we leverage optimal transport (OT) which has gained momentum in the machine learning community to propose a novel and scalable biclustering model that generalizes several classical biclustering approaches. We perform extensive experimentation to show the validity of our approach compared to other OT biclustering algorithms along both dimensions of the dyadic datasets.

## 1  Introduction

Let $G = (U, V, E)$ be a *bipartite graph*, which is a graph whose vertices can be divided into two disjoint sets $U = \{1, 2, \ldots, |U|\}$ with $|U| = n$, $V = \{1, 2, \ldots, |V|\}$ with $|V| = d$ and the set of edges $E$ where each edge connects a vertex of $U$ to a vertex of $V$. The adjacency matrix for this type of graph has the following structure

$$\mathbf{A} = \begin{pmatrix} \mathbf{0}_{n \times n} & \mathbf{B} \\ \mathbf{B}^\top & \mathbf{0}_{d \times d} \end{pmatrix} \tag{1}$$

where $\mathbf{B}$ of size $n \times d$ is called the *biadjacency matrix* of $G$, its rows and columns corresponding to the two sets of vertices; each entry represents an edge between a row and a column. *Biclustering* (or *Co-clustering*) is the extention of clustering to this type of graph. Following [21], several biclustering models have attempted to solve the problem by viewing $\mathbf{B}$ as a two-mode matrix and searching for a simultaneous partition of its rows and columns [9]. In this way, biclustering seeks to reveal subsets of $U$ which exhibit a similar behaviour across a subset of $V$ in matrix $\mathbf{B}$.

Biclustering has been used in a number of different contexts. [12] used microarray data to find relations between genes and conditions, finding that genes with similar functions often cluster together. [20] applied this paradigm to data from the US Food and Drug Administration reporting system in order to identify groups of drugs with adverse effects. [11] used it to find market segments among tourists so as to enable more effective targeted marketing. There have been various other applications [9, 33, 19].

Several solutions to the biclustering problem have been proposed in the literature (see [17]). [10] used an information-theoretic approach to solve the problem by minimizing the difference in mutual

36th Conference on Neural Information Processing Systems (NeurIPS 2022).

information between $\mathbf{B}$ and a summary matrix; they implicitly assume that the data points are generated from a Poisson latent block model [18]. [3] adapted classical modularity to bipartite networks and then used it to identify modules within them. [35] proposed a biclustering paradigm based on nonnegative matrix tri-factorization of the biadjacency matrix.

Recently, *Optimal Transport* (OT) has taken the machine learning community by storm. OT has helped to solve a variety of data mining problems, and biclustering is no exception. [25] proposed two models for biclustering: a first model, CCOT, which does co-clustering based on the scaling vectors obtained by applying the Sinkhorn-Knopp algorithm on a square subsampled version of matrix $\mathbf{B}$, and a second model, CCOT-GW, which uses scaling vectors obtained by computing entropic Gromov-Wasserstein barycenters, and which does not require subsampling. Then came [34], where the authors did biclustering by minimizing a new metric, COOT, which generalizes the Gromov-Wasserstein distance between $\mathbf{B}$ and a summary matrix, similarly to what was done in [10]. More specifically, they proposed two new metrics: COOT, together with an entropically regularized metric COOT$_\lambda$. However, both [25] and [34] have certain drawbacks. First, both algorithms do not tackle the biclustering from the beginning; the co-clusters are deduced at the convergence. Thereby biclustering is a consequence and not a main goal. Secondly, they suffer from high computational complexity; CCOT and CCOT-GW also consume large amounts of memory. Finally, we will see that these algorithms are not suited to dyadic sparse data.

In this paper, while integrating the biclustering objective from the beginning, we propose a generic framework for biclustering through optimal transport, which generalizes some previous biclustering approaches. We propose two efficient methods for solving this problem: one that gives an almost hard biclustering, and another that gives a *fuzzy* or *soft* biclustering through entropic regularization. These methods outperform other optimal transport biclustering models, in terms of both document and term clustering, on several regular and large scale datasets, while being more computationally and memory efficient. We emphasize once again that the approach we propose is specifically tailored to datasets consisting of dyadic data.

## 2 Methodology

**Notations.** In what follows, $\Delta^n = \{\mathbf{p} \in \mathbb{R}_+^n | \sum_{i=1}^n p_i = 1\}$ denotes the $n$-dimensional standard simplex. $\Pi(\mathbf{w}, \mathbf{v}) = \{\mathbf{Z} \in \mathbb{R}_+^{n \times k} | \mathbf{Z}\mathbf{1} = \mathbf{w}, \mathbf{Z}^\top \mathbf{1} = \mathbf{v}\}$ denotes the transportation polytope, where $\mathbf{w} \in \Delta^n$ and $\mathbf{v} \in \Delta^k$ are the marginals of the joint distribution $\mathbf{Z}$ and $\mathbf{1}_n$ is a vector of ones. Matrices are denoted with uppercase boldface letters, and vectors with lowercase boldface letters. For a matrix $\mathbf{M}$, its $i$-th row is $\mathbf{m}_i$ and its $j$-th column is $\mathbf{m}_j'$ We have that $\|.\|_0$ is the 0-norm which returns the number of nonzero elements of its argument.

### 2.1 Preliminaries

We first need to introduce exact discrete OT and its entropically regularized counterpart, and show how biclustering can be posed as an integer program.

**Discrete OT as a linear program.** The goal of discrete optimal transport is to find a minimal cost transport plan between a source probability distribution $\mathbf{w}$ and a target distribution $\mathbf{v}$. Here we are interested in the discrete case of the Kantorovich formulation of OT, that is

$$\text{OT}(\mathbf{M}, \mathbf{w}, \mathbf{v}) \triangleq \min_{\mathbf{Z} \in \Pi(\mathbf{w}, \mathbf{v})} \langle \mathbf{M}, \mathbf{Z} \rangle \tag{2}$$

where $\mathbf{M} \in \mathbb{R}^{n \times k}$ is the cost matrix, and $m_{ij}$ quantifies the effort needed to transport a probability mass from $\mathbf{w}_i$ to $\mathbf{v}_j$.

**Discrete entropy regularized OT.** It has been suggested in the literature [6, 5] that the use of a regularization such as entropic regularization can lead to better computational and statistical efficiency.

$$\text{OT}_\lambda(\mathbf{M}, \mathbf{w}, \mathbf{v}) \triangleq \min_{\mathbf{Z} \in \Pi(\mathbf{w}, \mathbf{v})} \langle \mathbf{M}, \mathbf{Z} \rangle - \lambda H(\mathbf{Z}) \tag{3}$$

where $H$ is the entropy defined as $H(\mathbf{Z}) \triangleq -\sum_{i,j} z_{ij} \log z_{ij}$ and $\lambda$ controls the strength of regularization. The computational efficiency comes from the fact that the unique solution of this problem is of the structure $\mathbf{Z} := \mathtt{diag}(\mathbf{a}) \exp(-\mathbf{M}/\lambda) \mathtt{diag}(\mathbf{b})$, a rescaled elementwise negative exponential of the cost $\mathbf{M}$, where $\mathbf{a}$ and $\mathbf{b}$ are scaling vectors. These vectors can be found efficiently using the Sinkhorn-Knopp algorithm.

**Biclustering as an integer program.** The *Block seriation* problem [27] consists in finding two permutation matrices, one for the rows and one for the columns s.t. dense blocks appear along the diagonal of the permuted matrix. A possible definition of the block seriation problem is as follows: given a matrix $\mathbf{B} \in \mathbb{R}^{n \times d}$ s.t $b_{ij}$ gives the strength of the association between row $i$ and column $j$ (such as in the case of a biadjacency matrix, for example), we have

$$\max_{\mathbf{C}} \quad \sum_{i,j} b_{ij} c_{ij} \tag{4}$$

$$\text{subject to} \quad \forall\, i,j \quad c_{ij} \in \{0,1\} \qquad \forall\, i,j,i',j' \quad \begin{aligned} c_{ij} + c_{ij'} + c_{i'j'} - c_{i'j} &\leq 2 \\ c_{i'j'} + c_{i'j} + c_{ij} - c_{ij'} &\leq 2 \\ c_{i'j} + c_{ij} + c_{ij'} - c_{i'j'} &\leq 2 \\ c_{ij'} + c_{i'j'} + c_{i'j} - c_{ij} &\leq 2 \end{aligned}$$

$$\forall\, j \quad \sum_i c_{ij} \geq 1$$

$$\forall\, i \quad \sum_j c_{ij} \geq 1$$

A solution $\mathbf{C}$ is a block diagonal matrix up to a permutation of its rows and columns. The block seriation problem is an integer programming problem that is NP-hard. One approach for solving this problem uses a simplified version where a rank constraint $\mathtt{rank}(\mathbf{C}) \leq k$ is added for $k$ the number of desired biclusters. Integrating this constraint into (4), we can define a new problem by low-rank factorization of $\mathbf{C}$, i.e. $\mathbf{C} = \mathbf{Z}\mathbf{W}^\top$, which we formulate as

$$\max_{\substack{\mathbf{Z} \in \Gamma(n,k) \\ \mathbf{W} \in \Gamma(d,k)}} \sum_{i,j,h} b_{ij} z_{ih} w_{jh} \tag{5}$$

where $\Gamma(n,k) = \{\mathbf{Z} \in \{0,1\}^{n \times k} \mid \mathbf{Z}\mathbf{1} = \mathbf{1}\}$ is the set of hard partitions of dimension $n \times k$. A simple heuristic for solving this problem involves alternatingly solving for $\mathbf{Z}$ given $\mathbf{W}$, and vice-versa, using classical clustering algorithms, before identifying biclusters through the rearranged matrix $\mathbf{C}$, which displays a block diagonal structure, as shown in figure 1a. The biclusters are identified by grouping together the rows and columns that form a block along the diagonal.

## 2.2 Biclustering using Optimal Transport

Here we propose a new biclustering problem based on block seriation and optimal transport. For this purpose we first define what we term an *anti-adjacency matrix*. Note that a similar concept has been discussed in [36].

**Definition 1 (Anti-adjacency matrix)** *Given a graph characterized by an adjacency matrix $\mathbf{A}$, we have a corresponding anti-adjacency matrix $\overline{\mathbf{A}}$ s.t. $\overline{a}_{ij}$ quantifies the discrepancy between nodes $i$ and $j$.*

We consider a bipartite graph characterized by its biadjacency matrix $\mathbf{B} = (b_{ij}) \in \mathbb{R}^{n \times d}$. The rows of $\mathbf{B}$ are endowed with weights $\mathbf{w} \in \Delta^n$ and its columns with weights $\mathbf{v} \in \Delta^d$. We also consider a row exemplar distribution $\mathbf{r} \in \Delta^r$ and a column exemplar distribution $\mathbf{c} \in \Delta^c$. Depending on the availability of *a priori* information about the data, these weight vectors can be set to uniform distributions.

Now let its anti-biadjacency matrix be $\overline{\mathbf{B}} = L(\mathbf{B})$, where $L : \mathbb{R}^{n \times d} \to \mathbb{R}^{n \times d}$ means that $b_{ij}$, the association between node $i$ and node $j$, is transformed into a discrepancy measure $L(\mathbf{B})_{ij}$. Thus, we define the optimal transport block seriation problem as the following bilinear program

$$\mathrm{BCOT}(\mathbf{w},\mathbf{v},\mathbf{r},\mathbf{c}) \triangleq \min_{\substack{\mathbf{Z} \in \Pi(\mathbf{w},\mathbf{r}) \\ \mathbf{W} \in \Pi(\mathbf{v},\mathbf{c})}} \sum_{i,j,k} L(\mathbf{B})_{ij} z_{ik} w_{jk} \equiv \min_{\substack{\mathbf{Z} \in \Pi(\mathbf{w},\mathbf{r}) \\ \mathbf{W} \in \Pi(\mathbf{v},\mathbf{c})}} \left\langle L(\mathbf{B}), \mathbf{Z}\mathbf{W}^\top \right\rangle \tag{6}$$

where $\mathbf{Z}$ is a transport plan (or coupling) between between the row distribution $\mathbf{w}$ and the row exemplar distribution $\mathbf{r}$, and similarly for $\mathbf{W}$ w.r.t. the column distribution $\mathbf{v}$ and the column exemplar distribution $\mathbf{c}$.

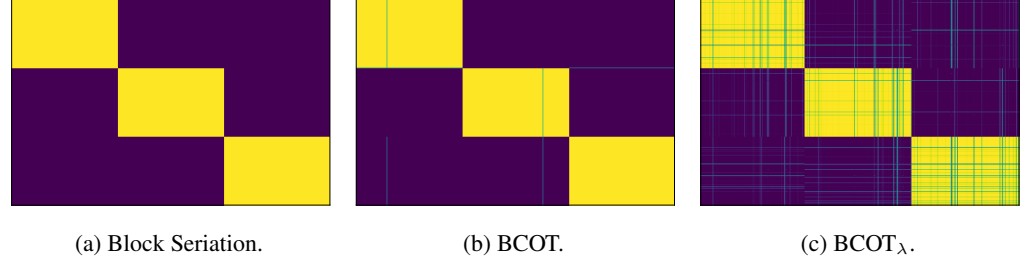

| (a) Block Seriation. | (b) BCOT. | (c) BCOT$_\lambda$. |

Figure 1: Biclusters formed using three different methods on the Pubmed dataset. Classical block seriation results in a biclustering that is hard. BCOT results in a biclustering that is almost hard with few nonzero entries outside the main block diagonal. BCOT$_\lambda$ results in a soft biclustering with many nonzero elements outside the block diagonal.

**Inducing a biclustering via BCOT.**   We will now show how to obtain a partition of the rows and the columns given a solution pair $(\mathbf{Z}, \mathbf{W})$. In what follows our aim is to identify an *almost-hard clustering* couple for rows and columns from the couplings $\mathbf{Z}$ and $\mathbf{W}$.

**Definition 2** (**$h$-almost hard clustering**) *We define an $h$-almost hard clustering as a clustering whose assignment matrix is $\mathbf{C} \in \mathbb{R}^{n \times k}$ s.t. $\|\mathbf{C}\|_0 = n + h$ and for each row $\mathbf{c}$ of $\mathbf{C}$ we have that $\|\mathbf{c}\|_0 > 0$. When $h = 0$, we obtain a standard hard clustering with one non-zero element per row.*

**Proposition 1** [1] *For $\mathbf{w}$, $\mathbf{v}$, $\mathbf{r}$ and $\mathbf{c}$ containing no zeros, there exists an optimal pair of coupling matrices $\mathbf{Z}$ and $\mathbf{W}$ that are $h$-almost hard clusterings with $h \in \{0, \dots, k-1\}$. Furthermore, when $n = k$ (resp. $d = k$) and $\mathbf{w} = \mathbf{r}$ (resp. $\mathbf{v} = \mathbf{c}$), this $\mathbf{Z}$ (resp. $\mathbf{W}$) becomes a hard clustering, i.e., $\mathbf{Z} \in \Gamma(n,n)$ (resp. $\mathbf{W} \in \Gamma(d,d)$).*

This means that the solutions are already almost a hard partition of the data, since $k << n, d$. To obtain a final hard clustering in the strict sense, we assign each row (resp. column) to the one corresponding to the row of $\mathbf{Z}$ (resp. $\mathbf{W}$) with the largest value. This should not significantly change the structure of the solution. Figure 1b provides an illustration: here we see the block diagonal structure generated by the product of the two coupling matrices $\mathbf{C} = \mathbf{Z}\mathbf{W}^\top$, with its similarity in appearance to the biclustering produced by the hard block seriation 1a, apart from a few nonzero entries off the block diagonal that are hard to see immediately.

**Intuition for BCOT.**   To explain the intuition behind the proposed approach we need to look at how the problem is solved. The optimization procedure as described in algorithm 1 consists in alternating between the computation of an optimal transport plan $\mathbf{Z}$ given $\mathbf{W}$ and vice versa. As regards solving for $\mathbf{Z}$ given $\mathbf{W}$, the problem can be rewritten as

$$\text{BCOT}(\mathbf{w}, \mathbf{v}, \mathbf{r}, \mathbf{c}) \equiv \min_{\mathbf{Z} \in \Pi(\mathbf{w}, \mathbf{r})} \langle L(\mathbf{B})\mathbf{W}, \mathbf{Z} \rangle . \tag{7}$$

This is an optimal transport problem with $L(\mathbf{B})\mathbf{W}$ as the cost matrix. The resulting transport plan $\mathbf{Z}$ can be seen as a kind of row cluster assignment matrix: if $z_{ih} > 0$, then row $i$ is assigned to cluster $h$. The same holds for $\mathbf{W}$, which can be seen as a column cluster assignment matrix. This also means that since $L(\mathbf{B})$ is the dissimilarity between the rows and the columns, then the cost matrix $L(\mathbf{B})\mathbf{W}$ represents the dissimilarity between rows and row exemplars (or representatives or centroids). In particular, $L(\mathbf{B})_i \mathbf{w}_h$ is the dissimilarity or cost of probability mass transportation between row $i$ and row cluster exemplar $h$. The reasoning is the same for the columns and the optimal coupling $\mathbf{W}$.

**Low-rank optimal transport.**   Biclustering is the main purpose of the approach we proposed, but there is another interesting use case.

**Proposition 2** *For equal target row and column representative distributions, i.e., $\mathbf{r} = \mathbf{c}$, and containing no zero entries, then given a solution pair $\mathbf{Z}$ and $\mathbf{W}$ to BCOT, the matrix $\mathbf{Q} = \mathbf{Z}\, diag(1/\mathbf{r})\mathbf{W}^\top$ is an approximation of the optimal transport plan that is a solution to problem (2) and whose rank is at most $\min(rank(\mathbf{Z}), rank(\mathbf{W}))$.*

---

[1]Proofs for the propositions are given in the appendix.

Some recent works [16, 31] have suggested that this kind of low-rank regularization is preferable to entropic regularization as regards certain aspects. For example, the rank parameter is easier to select, since it has simple bounds (an integer between $1$ and $n$). This may be contrasted with the regularization strength $\lambda$ in the Sinkhorn algorithm, which is continuous.

### 2.3 Fuzzy Biclustering via Regularized Optimal Transport

As previously mentioned, using entropic regularization may be interesting because of its various useful features including statistical and computational efficiency. However, another feature of entropic regularization is that the optimal couplings $\mathbf{Z}$ and $\mathbf{W}$ are dense matrices as a consequence of the structure of the optimal solution of entropically regularized OT problems. We formulate the problem as follows

$$\text{BCOT}_\lambda(\mathbf{w}, \mathbf{v}, \mathbf{r}, \mathbf{c}) \triangleq \min_{\substack{\mathbf{Z} \in \Pi(\mathbf{w},\mathbf{r}) \\ \mathbf{W} \in \Pi(\mathbf{v},\mathbf{c})}} \left\langle L(\mathbf{B}), \mathbf{Z}\mathbf{W}^\top \right\rangle - \lambda_{\mathbf{Z}} H(\mathbf{Z}) - \lambda_{\mathbf{W}} H(\mathbf{W}) \tag{8}$$

where $\lambda_{\mathbf{Z}}$ and $\lambda_{\mathbf{W}}$ are the regularization parameters.

**Fuzzy block seriation.** We propose a fuzzy variant of the block seriation problem that allows us by extension to define a fuzzy variant for BCOT using entropic regularization. Let the fuzzy block seriation problem be defined as

$$\max_{\substack{\mathbf{Z} \in \Gamma_s(n,k) \\ \mathbf{W} \in \Gamma_s(d,k)}} \sum_{i,j,h} b_{ij} z_{ih} w_{jh} + \Omega(\mathbf{Z}, \mathbf{W}) \tag{9}$$

where $\Omega(\mathbf{Z}, \mathbf{W})$ is some regularization term introduced to make the partition matrices $\mathbf{Z}$ and $\mathbf{W}$ dense (for example, entropic regularization or low-rank constraints), and $\Gamma_s(n,k) = \{\mathbf{Z} \in \mathbb{R}_+^{n \times k} | \mathbf{Z}\mathbf{1} = \mathbf{1}\}$ is the set of fuzzy partitions. Intuitively, for a solution pair $(\mathbf{Z}, \mathbf{W})$, up to a constant factor, each entry in the block seriation matrix $\mathbf{C} = \mathbf{Z}\mathbf{W}^\top$ can be seen as the probability of its corresponding row and column belonging to the same bicluster i.e. $c_{ij} = \mathbf{z}_i \mathbf{w}_j = \sum_{h=1}^r z_{ih} w_{jh} = p(\mathbf{b}_i, \mathbf{b}'_j) = \sum_{h=1}^r p(\mathbf{b}_i, \mathbf{b}'_j \in h)$.

It is easy to see how problem (9) is related to problem (8) and that the couplings corresponding to solutions of the problem give the probability that the different rows and columns belong to the same biclusters. Figure 1c shows biclusters produced by the solutions of $\text{BCOT}_\lambda$. Similarly to BCOT, a block diagonal structure is formed. However, there are also several off-block diagonal nonzero entries that represent the probabilities of the row-column pairs belonging to the same biclusters.

## 3 Links to Existing Work

### 3.1 Modularity Maximization in Bipartite Graphs [3].

This model is able to co-cluster binary and contingency matrices by directly maximizing an adapted version of the modularity measure traditionally used for networks. The criterion that it optimizes is

$$\max_{\substack{\mathbf{Z} \in \Gamma(n,k) \\ \mathbf{W} \in \Gamma(d,k)}} \sum_{i,j,h} z_{ih} w_{jh} \left( b_{ij} - \frac{b_{.j} b_{i.}}{b_{..}} \right). \tag{10}$$

By setting $L(\mathbf{B}) = -(\mathbf{B} - \frac{1}{b} \mathbf{B}\mathbf{1}\mathbf{1}^\top \mathbf{B})$, this problem becomes equivalent to ours; the difference is in the constraints on $\mathbf{Z}$ and $\mathbf{W}$.

### 3.2 Modularity-Based Sparse Soft Graph Clustering [23].

Here the authors proposed a fuzzy variant of the above problem (although in the context of traditional clustering rather than biclustering). Solving the problem gives, for each element of the dataset, a probability of that element belonging to a given cluster. Our proposed entropic regularization variant represents a kind of extension of this problem to bipartite graphs.

### 3.3 Directional Co-clustering with a Conscience [30, 1].

This model makes use of the block von Mises-Fisher mixture model for co-clustering directional data on the unit-sphere. It optimizes the following criterion:

$$\max_{\substack{\mathbf{Z}\in\Gamma(n,k)\\\mathbf{W}\in\Gamma(d,k)}} \sum_{i,j,h} \frac{1}{\sqrt{z_{.h}w_{.h}}} z_{ih}w_{jh}b_{ij}. \tag{11}$$

In our formulation, if we define $L(\mathbf{B}) = -\mathbf{B}$ and apply cluster size normalization on the optimal transport plans $\tilde{\mathbf{Z}} = \mathbf{Z}\texttt{diag}(\mathbf{Z}^\top \mathbf{1})^{-1/2}$ and $\tilde{\mathbf{W}} = \mathbf{W}\texttt{diag}(\mathbf{W}^\top \mathbf{1})^{-1/2}$ after computing $\mathbf{Z}$ and $\mathbf{W}$ respectively in algorithm 1, we obtain a more general version of the algorithm proposed by the authors for solving problem (11).

### 3.4 Bipartite Correlation Clustering [2].

In the case where the cost function results in a complete bipartite graph with '+' and '-' edges with a function

$$L(\mathbf{B})_{ij} = \begin{cases} -1 & \text{if } b_{ij} > 0 \\ +1 & \text{otherwise} \end{cases} \tag{12}$$

we get what is known as Bipartite Correlation Clustering. The solution to this problem maximizes the number of agreements, i.e. the number of all '+' edges within clusters plus all '-' edges distributed across clusters.

## 4 Optimization and Complexity

**Optimization.** Since the block seriation problem is NP-hard, computing an exact solution is prohibitive. An efficient and widely used heuristic for solving these kinds of problems involves the use of block coordinate descent, where row assignments are computed for fixed column assignments, and then vice versa, in alternation. We express the proposed algorithm in pseudo-code as algorithm 1. At each iteration we solve two intermediate optimal transport problems with cost matrices of dimensions $n \times k$ and $d \times k$, since $\mathbf{B}$ is generally sparse, and $L$ can be defined such that $L(\mathbf{B})$ retains a similarly sparse structure. The computation of the intermediate cost matrices $L(\mathbf{B})\mathbf{W}$ and $L(\mathbf{B})^\top \mathbf{Z}$ is reasonably efficient. We also observed that the algorithm does not need many iterations to converge, as shown in figure 2, be it for BCOT or BCOT$_\lambda$.

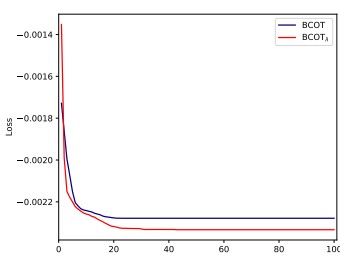

Figure 2: Loss for BCOT and BCOT$_\lambda$ on Pubmed.

---

**Algorithm 1:** BCOT

**Input** : $\mathbf{B}$ bi-adjacency matrix, $\mathbf{w}$ and $\mathbf{v}$ row and column weights, $\mathbf{r}$ and $\mathbf{c}$ row and column exemplar distributions
**Output :** $\pi^r$, $\pi^c$ row and column partitions
$\mathbf{W} \leftarrow \mathbf{W}_{init}$;
**while** *not converged* **do**
    $\mathbf{Z} \leftarrow \arg \texttt{OT}\left(L(\mathbf{B})\mathbf{W}, \mathbf{w}, \mathbf{r}\right)$;
    $\mathbf{W} \leftarrow \arg \texttt{OT}\left(L(\mathbf{B})^\top \mathbf{Z}, \mathbf{v}, \mathbf{c}\right)$;
**end**
Generate $\pi^r$, $\pi^c$ from $\mathbf{Z}$ and $\mathbf{W}$;

---

**Proposition 3** *The computational complexity of the* BCOT *algorithm 1 when using an exact OT solver is* $\mathcal{O}\left(tk\|\mathbf{B}\|_0 + tnk(n+k)\log(n+k) + tdk(d+k)\log(d+k))\right)$, *and when using entropic regularization the complexity is* $\mathcal{O}(tk\|\mathbf{B}\|_0 + tkn + tkd)$, *where t is the number of iterations.*

In table 1, we report the computational and spatial complexities of the different biclustering approaches. Our model has the same spatial complexity as the COOT variants and a better complexity

Table 1: Computational and spatial complexity of the different OT biclustering approaches. For COOT variants, we report complexities for an euclidean cost matrix. For a generic cost, the time complexity is greater. For simplicity, we suppose that $d \in O(n)$ and that we want a biclustering with the same number of row and column clusters for COOT and CCOT. $t$ denotes the number of iterations and for CCOT, $s$ denotes the number of necessary samplings.

| Method | Spatial complexity | Time complexity |
|---|---|---|
| CCOT | $O(n^2)$ | $O(sn^3)$ |
| CCOT-GW | $O(n^2)$ | $O(n^3)$ |
| COOT* | $O(nk)$ | $O((n+k)nk + k^2n + t(n+k)nk\log(n+k))$ |
| COOT$^*_\lambda$ | $O(nk)$ | $O((n+k)nk + k^2n + tnk)$ |
| BCOT | $O(nk)$ | $O(k\|\mathbf{B}\|_0 + t(n+k)nk\log(n+k))$ |
| BCOT$_\lambda$ | $O(nk)$ | $O(k\|\mathbf{B}\|_0 + tnk)$ |

than CCOT variants. As regards the computational complexity, our model should in most cases be faster with sparse data, and our experiments support this conjecture. For reproducibility, we publicly release our code [2].

## 5 Experiments

We ran experiments using term-document matrices. The benefit of using biclustering on this kind of data is that the resulting biclusters contain both documents and the words that characterize them, which is helpful in interpreting the clustering of the documents. Additional experiments over synthetic and gene expression data are available in the appendix.

### 5.1 Datasets

We evaluate BCOT in relation to six benchmark document-term datasets: ACM, DBLP, PubMed, Wiki, Ohscal, and 20 Newsgroups. Their characteristics are shown in Table 2. ACM, DBLP, Pubmed and Wiki are attributed networks from which we use only the node-level features that correspond to term-document matrices. We also selected the Ohscal collection and 20 Newsgroups as large-scale document-term matrices to serve as computational efficiency benchmarks.

Table 2: Characteristics of the datasets.

| Dataset | #Documents | #Terms | #Document clusters | Sparsity (%) |
|---|---|---|---|---|
| ACM [13] | 3025 | 1870 | 3 | 95.52 |
| DBLP [13] | 4057 | 334 | 4 | 96.4 |
| PubMed [32] | 19717 | 500 | 3 | 89.98 |
| Wiki [37] | 2405 | 4973 | 17 | 86.99 |
| Ohscal [22] | 11162 | 11465 | 10 | 99.47 |
| 20 Newsgroups [26] | 18846 | 14390 | 20 | 99.41 |

### 5.2 Experimental Setup

In our experiments we define the loss function as $L(\mathbf{B}) = -c\mathbf{B}$, where $c$ is selected from $\{1, k, d, n\}$. For BCOT$\lambda$, the regularization parameter lambda is selected from $\{10^{-4}, 10^{-3}, 10^{-2}, 10^{-1}, 1, 10\}$. The best hyper-parameters are those that minimize the number of empty clusters. In the case of ties, we select according to the value of the Davies-Bouldin index of the partition [7]. Random restarts are not used for any of the algorithms, including $k$-means. We use the implementation provided by the authors for CCOT, CCOT$_\lambda$ and CCOT-GW. The code for CCOT was not available, and so we had to implement it based on the code for CCOT-GW. All the reported figures are the averages of 10 runs.

---

[2] https://github.com/chakib401/BCOT

All the experiments were performed on the same machine with an Intel(R) Xeon(R) CPU and 12GB RAM. For OT solvers we made use of the POT package [15].

## 5.3 Document Clustering

**Metrics.**    Here, the evaluation is straightforward, we adopt three popular clustering metrics: clustering accuracy (CA), normalized mutual information (NMI) [4], adjusted rand index (ARI) [24].

Table 3: Document clustering performance on the four datasets. OOM denotes out of memory.

| Method | ACM | | | DBLP | | | PubMed | | | Wiki | | |
|---|---|---|---|---|---|---|---|---|---|---|---|---|
| | CA | NMI | ARI | CA | NMI | ARI | CA | NMI | ARI | CA | NMI | ARI |
| $k$-Means | 51.1±11.3 | 13.7±11.2 | 14.0±10.6 | 36.9±2.4 | 10.4±2.0 | 4.3±2.0 | 52.3±4.7 | 18.2±10.5 | 15.3±10.1 | 26.0±6.1 | 18.6±9.3 | 3.3±2.9 |
| CCOT | 12.4±2.0 | 1.0±0.2 | 0.4±0.2 | 28.6±0.5 | 0.6±0.0 | 0.4±0.0 | 32.7±0.2 | 3.0±0.0 | 3.1±0.1 | 10.6±0.5 | 4.9±0.1 | 0.6±0.15 |
| CCOT-GW | 8.1±0.0 | 1.5±0.0 | 0.3±0.0 | 9.4±0.0 | 1.7±0.0 | 0.3±0.0 | | OOM | | 10.9±0.0 | 4.3±0.0 | 0.48±0.0 |
| COOT* | 39.0±0.0 | 1.9±0.0 | 2.0±0.0 | 30.5±1.4 | 1.4±0.3 | 1.2±0.3 | 43.2±1.5 | 1.7±0.6 | 1.3±1.5 | 25.9±1.8 | 28.7±2.2 | 12.3±1.7 |
| COOT$_\lambda$ | 41.5±0.2 | 1.9±0.1 | 2.2±0.0 | 30.6±0.0 | 0.7±0.0 | 0.6±0.0 | 42.4±1.5 | 1.7±0.5 | 1.0±1.3 | 17.2±0.0 | 1.7±0.0 | 0.31±0.0 |
| BCOT | **76.6±1.5** | **38.3±2.2** | **43.3±2.6** | **61.5±6.2** | **27.4±4.3** | **28.3±5.5** | 53.6±4.5 | 15.9±1.9 | 12.9±2.4 | 49.8±1.5 | 47.9±1.0 | 30.6±1.0 |
| BCOT$_\lambda$ | 76.2±0.6 | 37.6±0.8 | 42.4±1.0 | 59.4±9.9 | 26.6±7.6 | 27.2±9.5 | **56.5±3.1** | **18.4±1.3** | **15.4±1.8** | **50.8±1.5** | **49.4±0.9** | **31.9±0.8** |

**Performance.**    Document clustering results on ACM, DBLP, PubMed and Wiki are given in table 3 for the three metrics. In all cases the best result is obtained either by BCOT or by BCOT$_\lambda$. Moreover, on Wiki, BCOT$_\lambda$ gives competitive results when compared with state-of-the-art attributed graph clustering methods presented in [14], despite not having access to the graph structure information in the Wiki citation network.

**Efficiency.**    Figure 3 plots the document clustering performance (accuracy against training time) of the different methods on the two large-scale document-term matrices 20 Newsgroup and Ohscal. BCOT offers the best accuracy while BCOT$_\lambda$ is fastest method on both datasets. We see that for both BCOT and COOT, the entropic-regularized versions outspeed their exact counterparts and that CCOT suffers from very high computation times, due mainly to the fact that this method requires pairwise distance matrices to be computed on the rows and columns.

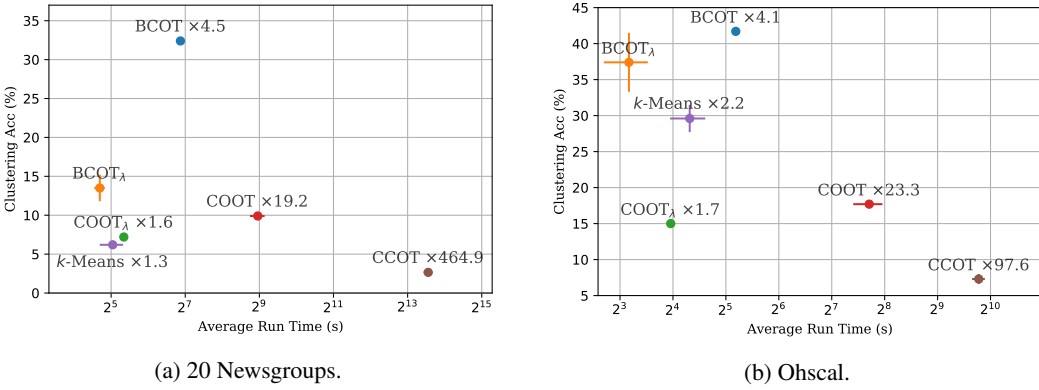

|                        |                |
|:----------------------:|:--------------:|
| (a) 20 Newsgroups.     | (b) Ohscal.    |

Figure 3: Accuracy against training time on NG20 and Ohscal. BCOT$_\lambda$ is the fastest and has a competitive level of accuracy. BCOT gives the best accuracy while remaining relatively efficient. The multiplication factors shown for the training times take BCOT$_\lambda$ as the reference (and so, for example, ×4.5 shown for BCOT means that it is approximately 4.5 times slower than BCOT$_\lambda$). We were not able to benchmark CCOT-GW since it failed to scale to these datasets.

## 5.4 Term Clustering

**Metrics.**    Unlike document clustering, there is no ground truth partition for terms, so we need to find another way of evaluating term clustering results. One generally acceptable technique is to analyse the semantic coherence of the clusters obtained. To this end we introduce a metric

based on *point mutual information* (PMI). PMI is a frequently used information-theoretic metric for quantifying the relationship between pairs of discrete random variable outcomes. The PMI measure was chosen because prior research [29] has shown that it is closely associated with human judgements in determining word relatedness. The PMI between the terms $w_i$ and $w_j$ is calculated as

$$\text{PMI}(w_i, w_j) = \log \frac{p(w_i, w_j)}{p(w_i)p(w_j)} \tag{13}$$

In the context of term clustering, given the word co-occurrence matrix $\mathbf{K} = \mathbf{B}^\top \mathbf{B}$, the PMI is estimated as in

$$\text{PMI}(w_i, w_j) = \log \frac{k_{..}k_{ij}}{k_{i.}k_{.j}} \tag{14}$$

To evaluate a partition of terms $\mathcal{P}$, we propose a metric based on *intra* and *inter* PMI metrics as follows:

$$\text{PMI}_{intra}(P) = \sum_{i \in P} \sum_{j \in P} k_{ij} \quad (15) \qquad\qquad \text{PMI}_{inter}(P) = \sum_{i \in P} \sum_{j \notin P} k_{ij} \quad (16)$$

In this way, a good clustering should reveal a high intra-cluster semantic relatedness, corresponding to higher PMI values. Using the *intra* and *inter* PMIs, we propose the following *coherence* index

$$\text{coherence}(\mathcal{P}) = \frac{1}{\sum\limits_{P \in \mathcal{P}} |P|} \sum_{P \in \mathcal{P}} |P| \left( \text{PMI}_{intra}(P) - \text{PMI}_{inter}(P) \right). \tag{17}$$

Our reasoning is this: the greater the semantic proximity between terms in the same clusters, and the greater the sematic distance between terms in different clusters, the higher the value of *coherence*.

**Results.** Since there is no ground truth number of term clusters, we use the cluster number estimations produced by CCOT-GW for all the other models so that it is easy to compare coherence values between them. Comparisons based on different numbers of clusters would favor the model using the larger number of clusters. Table 4 shows the coherences obtained across the different datasets using our approach, along with those of the baselines. It is clear that BCOT succeeds in capturing more semantics than the other approaches since, whatever the dataset, one or other of the two BCOT variants gives the highest coherence.

Table 4: Term clustering performance on the four datasets. OOM denotes out of memory.

| Method | ACM | DBLP | PubMed | Wiki | Ng20 | Ohscal |
|---|---|---|---|---|---|---|
| $k$-Means | 0.19±0.01 | 0.05±0.03 | 0.31±0.18 | 0.28±0.02 | 0.28±0.04 | 0.01±0.02 |
| CCOT | 0.03±0.00 | -0.07±0.06 | 0.02±0.01 | 0.02±0.00 | 0.05±0.00 | 0.06±0.00 |
| CCOT-GW | 0.08±0.00 | 0.03±0.00 | OOM | 0.01±0.00 | OOM | OOM |
| COOT | 0.12±0.01 | 0.07±0.00 | 0.14±0.01 | 0.40±0.00 | 0.43±0.02 | 0.23±0.01 |
| COOT$_\lambda$ | 0.21±0.00 | 0.04±0.00 | -0.00±0.00 | -0.08±0.00 | -0.02±0.00 | -0.13±0.00 |
| BCOT | **0.27±0.01** | **0.22±0.04** | 0.54±0.03 | **0.64±0.01** | **0.79±0.01** | **0.44±0.00** |
| BCOT$_\lambda$ | 0.24±0.00 | 0.16±0.02 | **0.57±0.01** | 0.62±0.01 | 0.27±0.01 | 0.35±0.00 |

## 5.5 Statistical Significance

We performed a Nemenyi post-hoc test [28, 8] with a confidence level of 90% on the document and term clustering results that we obtained, to determine whether our model outperforms other OT biclustering approaches in a statistically significant way. To conduct this test we generated 20 performance rankings of the OT biclustering models based on their performance for each dataset and quality metric pair for both document and term clustering. Figure 4 shows the results of the test. We see that two differently performing groups were identified, one comprising BCOT and BCOT$_\lambda$ and giving better results than the other group comprising the remaining COOT and CCOT variants. This indicates that with this specific number of datasets and metrics the test was unable to tell COOT and CCOT apart in a statistically significant way.

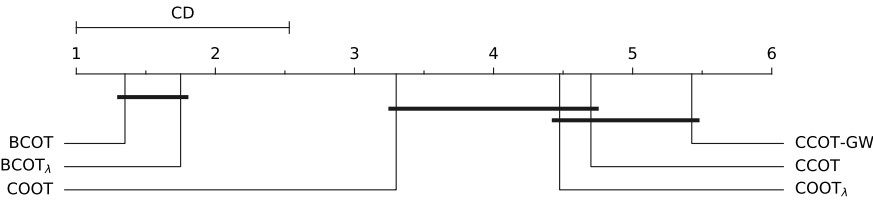

Figure 4: Result of the Nemenyi post hoc test.

## 6 Conclusion

Clustering and biclustering through optimal transport is still at a nascent stage, with many challenges remaining unsolved. This paper introduces a novel problem for biclustering using optimal transport that takes into account the sparse nature of certain types of dyadic data such document-term matrices, to enable more computationally efficient resolution. The problem is posed as a bilinear program that we solve using an efficient block coordinate descent algorithm to find a vertex solution. Experiments on a number of document-term datasets suggest that the proposed approach does a good job in finding clusters that correspond to ground truth document classes, while generating semantically coherent partitions for the terms. In this setting, our model outperforms recent OT biclustering methods by a significant margin, while being more computationally efficient.

## Acknowledgments and Disclosure of Funding

This work has been funded by Informatique Caisse des Dépôts et Consignations (ICDC), Association Nationale de la Recherche et de la Technologie (ANRT), and Idex-Spectrans of Université Paris Cité.

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
