# Efficient and Effective Optimal Transport-Based Biclustering: Supplementary Material

**Chakib Fettal**
Centre Borelli UMR 9010
Université Paris Cité
Informatique Caisse des Dépôts et Consignations
chakib.fettal@etu.u-paris.fr

**Lazhar Labiod**
Centre Borelli UMR 9010
Université Paris Cité
lazhar.labiod@u-paris.fr

**Mohamed Nadif**
Centre Borelli UMR 9010
Université Paris Cité
mohamed.nadif@u-paris.fr

## Appendix A   Proofs

***Proposition 1.*** For $\mathbf{w}$, $\mathbf{v}$, $\mathbf{r}$ and $\mathbf{c}$ containing no zeros, the resulting optimal coupling matrices $\mathbf{Z}$ and $\mathbf{W}$ are always an $h$-almost hard clustering with $h \in \{0, \ldots, k-1\}$. Furthermore, when $n = k$ (resp. $d = k$) and $\mathbf{w} = \mathbf{r}$ (resp. $\mathbf{v} = \mathbf{c}$), $\mathbf{Z}$ (resp. $\mathbf{W}$) represents a hard clustering $\mathbf{Z} \in \Gamma(n, n)$ (resp. $\mathbf{W} \in \Gamma(d, d)$).

**Proof for *proposition 1*.** The Kantorovich OT problem is a bounded linear program since $\Pi(\mathbf{w}, \mathbf{v})$ is a polytope i.e. a bounded polyhedron. The fundamental theorem of linear programming states that if the feasible set is non-empty then the solution lies in the extremity of the feasible region. This means that a solution $\mathbf{Z}$ to the OT problem is an extreme point of $\Pi(\mathbf{w}, \mathbf{v})$. We have that the extreme points of $\Pi(\mathbf{w}, \mathbf{v})$ can have at most $n + d - 1$ nonzero elements. To prove this we have to show that the bipartite graph induced by biadjacency matrix $\mathbf{Z}$, the solution to the optimal transport problem has no cycles. The maximum number of edges in an acyclic graph is $|V| - 1$ where $|V|$ is the number of nodes in the graph. Since the number of edges in the bipartite graph induced by biadjacency matrix $\mathbf{Z}$ is $n + d - 1$, the matrix $\mathbf{Z}$ can not have more than $n + d - 1$ nonzero entries. For a detailed proof see proposition 3.3 in [6].

We also have to show that for probability measures $\mathbf{w}$ and $\mathbf{v}$ that have no zero probability events, there is at minimum $\max(n, d)$ number of nonzero elements in $\mathbf{Z}$. This is straightforward since $\mathbf{w}$ and $\mathbf{v}$ contain no zeros, there will always be at least one nonzero element in every row and column of $\mathbf{Z}$ that represents some transfer of mass between elements of $\mathbf{w}$ and $\mathbf{v}$.

BCOT is a bilinear program that has a finite global solution which means that there exists at least one optimal solution pair $(\mathbf{Z}, \mathbf{W})$ such that $\mathbf{Z}$ is an extreme point of $\Pi(\mathbf{w}, \mathbf{r})$ and $\mathbf{W}$ is an extreme point of $\Pi(\mathbf{v}, \mathbf{c})$ (theorem 1 in [3]).

We then have that, For BCOT, $\mathbf{Z}$ has at most $n + k - 1$ and at least $\max(n, k) = n$ nonzero entries and that $\mathbf{W}$ has at most $d + k - 1$ and at least $\max(d, k) = d$ elements which are both $h$-almost hard clusterings with $h \in \{0, \ldots, k-1\}$.

When $n = k$ and $\mathbf{w} = \mathbf{r}$, the solution $\mathbf{Z}$ is a permutation matrix (up to a constant factor) and the number of nonzero elements in it is exactly $n$ which means that it represents a hard partition

36th Conference on Neural Information Processing Systems (NeurIPS 2022).

$\mathbf{Z} \in \Gamma(n, n)$. The proof is the same for $\mathbf{W}$. □

**Proposition 2.** Suppose that the target row and column representative distributions are the same, i.e. $\mathbf{r} = \mathbf{c}$ with no zero entries. Then, given a solution pair $\mathbf{Z}$ and $\mathbf{W}$ to BCOT, the matrix $\mathbf{Q} = \mathbf{Z}\,\mathtt{diag}(1/\mathbf{r})\mathbf{W}^\top$ is an approximation of the optimal transport plan that is a solution to problem The the Kantorovich OT problem and whose rank is at most $\min(\mathtt{rank}(\mathbf{Z}), \mathtt{rank}(\mathbf{W}))$.

**Proof of *proposition 2.*** From linear algebra, we have that $rank(\mathbf{Q}) \leq \min(rank(\mathbf{Z}), rank(\mathtt{diag}(1/\mathbf{r})), rank(\mathbf{W}))$. Since $\mathbf{Z}$ and $\mathbf{W}$ cannot have a rank greater than $k$ due to their dimension, and since $\mathtt{diag}(1/\mathbf{r})$ is a full rank matrix due to the assumption that $\mathbf{r}$ has no zero entries, we then have that $rank(\mathbf{Q}) \leq \min(rank(\mathbf{Z}), rank(\mathbf{W}))$.

For a proof that $\mathbf{Q}$ is indeed a valid transport plan i.e. $\mathbf{Q} \in \Pi(\mathbf{w}, \mathbf{v})$, we refer the reader to proposition 2.2 in [6]. □

**Proposition 3.** The computational complexity of the BCOT algorithm when using an exact OT solver is $\mathcal{O}\left(tk\|\mathbf{B}\|_0 + tnk(n + k)\log(n + k) + tdk(d + k)\log(d + k))\right)$, and when using entropic regularization the complexity is $\mathcal{O}(tk\|\mathbf{B}\|_0 + tkn + tkd)$, where $t$ is the number of iterations.

**Proof of *proposition 3.*** We suppose that $L(\mathbf{B})$ is a sparse matrix with the same number of nonzero entries as $\mathbf{B}$. The complexity of computing $L(\mathbf{B})\mathbf{W}$ and $L(\mathbf{B})\mathbf{W}$ in the BCOT algorithm is $\mathcal{O}(k\|\mathbf{B}\|_0)$.

The optimal transport problem can be formulated and solved as the Earth Mover's Distance (EMD) problem using any minimum-cost flow problem algorithm, such as one of the many variants of the network simplex algorithm. The authors in [5] proposed an algorithm for the network simplex in $\mathcal{O}(|V||E|\log|V|)$, where $|V|$ is the number of nodes and $|E|$ is the number of edges in the network. In our case, when solving the EMD for $\mathbf{Z}$ and cost matrix $L(\mathbf{B})\mathbf{W}$, the number of nodes is $|V| = n + k$ and the number of edges is $|E| = nk$, which means that the complexity of the operation is $\mathcal{O}(nk(n + k)\log(n + k))$. When computing the optimal transport plan for $\mathbf{W}$, for cost matrix $L(\mathbf{B})^\top\mathbf{Z}$, the complexity is $\mathcal{O}(dk(d + k)\log(d + k))$. The overall complexity of the BCOT algorithm is then $\mathcal{O}(k\|\mathbf{B}\|_0) + tnk(n + k)\log(n + k) + tdk(d + k)\log(d + k))$

When using entropic regularization the complexity is smaller, since computing the optimal transport plan requires only a transformation of the inputs matrix, which takes $\mathcal{O}(nk)$ in the $\mathbf{Z}$ computation step and $\mathcal{O}(dk)$ for $\mathbf{W}$. The ensuing application of the Sinkhorn-Knopp algorithm on the transformed matrices has complexities $\mathcal{O}(tnk)$ and $\mathcal{O}(tdk)$ for $\mathbf{Z}$ and $\mathbf{W}$ respectively, where $t$ is the number of iterations necessary. The overall complexity of BCOT$_\lambda$ is then $\mathcal{O}(k\|\mathbf{B}\|_0) + tnk + tdk)$, here $t$ includes the number of iterations of our algorithm as well as that of Sinkhorn-Knopp. □

## Appendix B   Additional Experiments

### B.1   Experiments on Synthetic Data

**Datasets.** As datasets with labels along both rows and columns are unavailable, we use synthetic data as in [4, 7]. Their structure is shown in figure 1, while their characteristics are reported in table 1.

Table 1: Characteristics of the synthetic datasets.

|   | Rows | Cols | Biclusters | Sizes | Sparse | Structure |
|---|------|------|------------|-------|--------|-----------|
| A | 500  | 500  | 10         | equal   | Yes | Block diagonal |
| B | 800  | 1000 | 6          | unequal | No  | Block diagonal |
| C | 800  | 800  | 7          | equal   | No  | Checkerboard |
| D | 2000 | 1200 | 4          | unequal | No  | Checkerboard |

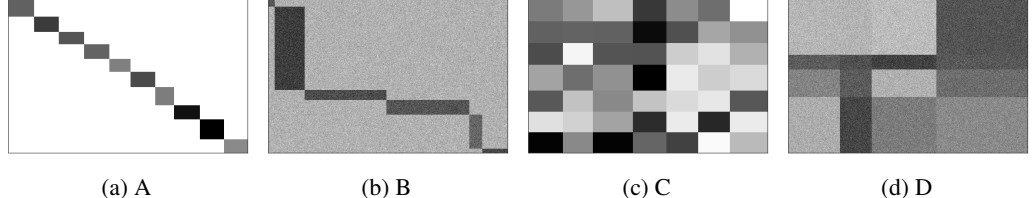

| (a) A | (b) B | (c) C | (d) D |

Figure 1: Synthetic datasets rearranged with respect to the true partition.

**Metrics.** From row $\pi^r$ and column $\pi^c$ clusters, we use the *Co-Clustering Accuracy (CCA)* proposed by [2] to compare two pairs of partitions. CCA is defined from *Clustering Accuracy* (CA) associated to $\pi^r$ and $\pi^c$ in comparison with the true row and column clusters; it is given by

$$\mathrm{CCA}(\pi^r, \pi^c) = \mathrm{CA}(\pi^r) + \mathrm{CA}(\pi^c) - \mathrm{CA}(\pi^r) \times \mathrm{CA}(\pi^c).$$

**Results.** We report the biclustering performance on the synthetic datasets in table 2. At least one of our models finds the perfect partition in all cases. These tests additionally allow us to show the utility of the the row cluster distribution **r** and column cluster distribution **c**. The use of these ground truth distributions resulted in an increase of 19.5 and 4.2 points for BCOT on C and D, and an increase of 0.3 and decrease of 0.8 for $\mathrm{BCOT}_\lambda$ on C and D.

Table 2: Biclustering performance on four synthetic datasets. gnd stands for ground truth.

| Method | A | B | C | D |
|---|---|---|---|---|
| $k$-means | **100.0±0.0** | 95.0±5.0 | 95.3±4.0 | 96.6±4.7 |
| CCOT | 54.4±3.5 | 70.0±0.0 | 29.7±0.4 | 55.7±1.8 |
| CCOT-GW | 99.1±0.0 | 83.5±0.0 | 83.4±0.0 | 75.3±0.0 |
| COOT | 99.8±0.0 | 78.8±2.0 | 99.8±0.0 | 93.7±1.2 |
| $\mathrm{COOT}_\lambda$ | 39.9±2.4 | 84.9±4.6 | 28.2±0.0 | 60.7±0.0 |
| BCOT | 99.8±0.0 | 80.4±2.2 | 99.6±0.1 | 91.3±0.7 |
| $\mathrm{BCOT}_\lambda$ | **100.0±0.0** | 99.1±0.4 | **100.0±0.0** | **100.0±0.0** |
| BCOT (gnd **r**, **c**) | same **r**, **c** | 99.9±0.0 | same **r**, **c** | 95.5±2.3 |
| $\mathrm{BCOT}_\lambda$ (gnd **r**, **c**) | same **r**, **c** | **100.0±0.0** | same **r**, **c** | 99.2±0.9 |

### B.2 Experiments on Gene Expression Data

**Datasets.** The gene-expression matrices used are the Cumida Breast Cancer and Leukemia datasets. Their characteristics are shown in Table 3.

Table 3: Characteristics of the gene expression datasets.

| Dataset | Samples | Genes | $k$ | Sparsity (%) |
|---|---|---|---|---|
| Breast Cancer [1] | 26 | 42945 | 2 | 0.0 |
| Leukemia [1] | 64 | 22283 | 5 | 0.0 |

**Metrics.** The metrics are the same as for document clustering.

**Performance** In table 4, we report results on the two micro-array datasets, $\mathrm{BCOT}_\lambda$ has the best performance on both of them.

Table 4: Gene clustering performance on the two microarray datasets.

| Method | Breast Cancer | | | Leukemia | | |
|---|---|---|---|---|---|---|
| | CA | NMI | ARI | CA | NMI | ARI |
| $k$-means | 75.8±18.0 | 41.9±40.5 | 31.2±49.0 | 74.8±7.2 | 72.1±5.4 | 50.1±8.3 |
| CCOT | | OOM | | 40.6±0.0 | 0.0±0.0 | 0.0±0.0 |
| CCOT-GW | | OOM | | | OOM | |
| COOT | 63.1±5.2 | 5.4±8.7 | -1.2±2.9 | 36.2±2.7 | 14.0±3.6 | 5.4±3.2 |
| COOT$_\lambda$ | 61.5±0.0 | 5.4±0.0 | 2.2±0.0 | 32.5±3.3 | 8.7±2.7 | -.5±2.1 |
| BCOT | 76.9±0.0 | 37.2±0.0 | 26.7±0.0 | 71.2±5.4 | 59.6±6.9 | 39.9±6.3 |
| BCOT$_\lambda$ | **84.6±0.0** | **48.3±0.0** | **46.0±0.0** | **80.9±3.8** | **70.9±4.1** | **55.3±3.3** |