# OpenReview forum: "Efficient and Effective Optimal Transport-Based Biclustering"
_NeurIPS.cc/2022/Conference — NeurIPS 2022 Accept_

### Official Review · Reviewer_c4gm · 2022-07-11

**Rating:** 7
**Confidence:** 3
**Soundness:** 3 good
**Presentation:** 3 good
**Contribution:** 3 good

**Summary:**

The paper proposes a new approach to bi-clustering, BICOT, and its entropic regularization BICOT_{\lambda}. The proposed BICOT  model is quite general. In primis, it can be reduced to a low-rank solution for optimal transport under specific distribution conditions. Secondly, it can be reduced to several bi-clustering models in the literature.

The authors also prove that the computational complexity of BICOT is the same as that of previous Co-Clustering work COOT [26] and inferior to the versions of CCOT [19]. To support their claims, parallel to the theoretical demonstrations, the authors conduct an accurate analysis of their algorithm by comparing their method with state of the art for different dyadic data sets with increasing size. In particular, besides better accuracy, adjusted random index and normalized mutual information, BICOT_{\lambda} performs best in all the experiments.

From a theoretical point of view, this work introduces the problem in a more general and straightforward form that can be traced back through appropriate choices to several models previously introduced in the literature. The main idea is to formulate the bi-clustering problem by using two building blocks: block seriation and optimal transport.


**Questions:**

-  In (9) L is not used; it is not well explained in which sense (9) is related to (8).
-  In [26] is proved that COOT is a distance. Some discussion comparing L in COOT and BICOT would be appreciated. Comparisons with [26] are done on documents and terms, and in the paper it is said that it would not be possible on images. On the other hand, COOT has shown experiments on MNIST and USPS (see Figure 1 in [26] (on Neurips)), simply normalizing pixel magnitude to [0,1].
Computational complexity of [26] is O(min{(n + n’)dd’ + n’^2n; (d + d’)nn’ + d’^2d})
-    The performance of the proposed approach strongly relies on the choice of the function $L$. How much will the results change when changing $L$?
-    Is there a way to define an optimal $L$ for a given quality measure?
-    How the algorithm asymptotic complexity will change if one uses a dense function $L$?
-    How much do the results depend on the regularization function $\Omega$ in $BCOT_{\lambda}$?
-    How have been chosen the hyper-parameters of k-means clustering?

Typos:
 -Line 99:  L(b)_{ij}    remove Z
 -Line 101: two times “between”
- (12) L(b_{ij})
- Line 191: in --> is
- Line 194 is available
 -Line 280: till—>still

Proof of proposition 3:
Line 445 twice L(B)W -->  L(B)W and L(B)^TZ


**Limitations:**

The limitations of the proposed approach are not fully explained. For example, it seems that the asymptotic low cost of the algorithm is largely due to the sparseness of the cost matrix. Although one has the freedom to choose this matrix, for some applications a sparse matrix may not give satisfactory results. However, the use of dense matrices could considerably slow down the algorithm making it de facto uncompetitive.

**Strengths And Weaknesses:**

The strength of this work is mainly based on three aspects:
1.	The proposed approach generalizes many already present algorithms that can be obtained as special cases.
2.	The presented algorithm is more efficient both in terms of complexity and in terms of memory usage.
3.	The authors provide evidence of the effectiveness of their approach both theoretically and experimentally.
The paper conducts extended analysis over several data-set showing great performance compared to other existing approaches. The results are well exposed, and the paper is well written and easy to follow.

A possible weakness is that determining the number of clusters requires resorting to external metrics. Furthermore, it is unclear if the authors set the hyper-parameters by performing cross-validation on a data set independently from the validation set. Also, there is no reference to the hyper-parameters used in the methods that serve as the baseline. Finally, concerning the term clustering experiment, it seems that the best performances are due to the fact that some methods have not been optimized.

---

> ### Author Response · Authors · 2022-08-02
> **Some details on BCOT**
>
> Thank you for your thorough review and for supporting our proposal. We appreciate the comments and suggestions that we are willing to address in the revised version. A first revised version is already available.
>
> # Addressing the perceived weaknesses
>
> 1. Since we are in an unsupervised context, the ground truth labels are not available during training, so there is no concept of cross-validation or validation set. We tune the hyper-parameters according to rules of thumb and internal unsupervised metrics such as the Davies-Bouldin index which we used in our case.
>
> 2. There were not many parameters to tune in the baseline models, we used the ground truth number of clusters for all baseline. For the rest of the hyper-parameters, we used the default values prescribed by the authors of each baseline as there were no heuristics on how to tune specific hyper-parameters.
>
> # Answers to questions
>
> 1. They are different objective functions, BCOT$_\lambda$ (8) is to the fuzzy block seriation problem (9) what BCOT (6) is to the block seriation problem (5). We introduced (9) due the fact that there was no prior concept of a fuzzy block seriation in literature (to the best of our knowledge).
>
> 2. BCOT can be reduced to a distance, but only when a set of very restrictive conditions are respected, i.e., n=d=k, $\mathbf{r}=\mathbf{c}$ and $L(B)$ being a distance matrix. With this we recover the classical definition of the Wasserstein metric with BCOT and the sinkhorn distance with BCOT$_\lambda$ and lose the clustering component of our proposition. Concerning applications on images, sorry the sentence was incomplete in the sense that this requires preprocessing as you have suggested.
>
> 3. The choice of $L$ proposed in our contribution appears efficient and effective in terms of co-clustering. As we saw in section 3, other choices of $L$ leading to other co-clustering criteria and algorithms would be interesting to investigate, for instance, in an ensemble approach.
>
> 4. We have proposed a simple rule of thumb to see if a chosen $L$ function is a good candidate by looking at the number of clusters retrieved (no empty clusters).
>
> 5. When $B$ is dense then $\Vert B \Vert_0=nd$, we obtain the new computational complexity by replacing $\Vert B \Vert_0$ with $nd$ in **table 1**. The term $k\Vert B \Vert_0$ becomes $knd$ which is just the complexity of computing $L(B)^\top Z$ and $L(B)W$ (given that we assumed that n=d for simplicity).
>
> 6. The regularization function in BCOT$_\lambda$ is the entropy function. We have not tried other regularizations. In (9), we use $\Omega$ to propose a fuzzy variant of the block seriation problem.
>
> 7.The only hyper-parameter we set for $k$-means is the number of clusters for which we assume that the ground truth number is given.

---

### Official Review · Reviewer_Piwj · 2022-07-18

**Rating:** 6
**Confidence:** 4
**Soundness:** 3 good
**Presentation:** 2 fair
**Contribution:** 3 good

**Summary:**

This paper proposes a generic framework for biclustering using optimal transport. Two methods are developed in this framework and usually result in an almost hard biclustering and a fuzzy biclustering accordingly. The computational efficiency and accuracy are validated through six benchmark datasets.

**Questions:**

1. The introduction of the relevant is unclear and would need more details and formalization. I fail to grasp exactly what causes the weakness of the these introduced methods CCOT, CCOT-GW, COOT and COOT-GW.

2. When the authors say integrating the constraint $rank(C) \leq k$ to (4) and finally get (5),
are the three constraints (binarity, assignment, impossible triads) included or ignored?

3. The definition of anti-adjacency matrix seems a little arbitrary since it is defined based on the “discrepancy” between two nodes, which seems to be a concept without any rigorous mathematical statement in the paper.

4. Page 3 line 95. It seems that the dimension of r and c and be different. How can that be achieved considering Z should be a matrix with n row r columns and W should be a matrix with d row and c columns.

5. It would be better to replace the notation for the assignment matrix since it is duplicated with the solution matrix C. This causes some confusions during the reading.

6. Page 5, is there a connection between (8) and (9) or they are two different objective functions?

7. Page 7. The time complexity of CCOT reported in the paper “Coclustering through optimal transport” seems do not match with the order in the Table 1.  What causes the extra computational burden in this paper?

8. In the implementation, it seems the algorithm needs r and c as input. The details about how to choose these two parameters and decide the rank k seems to be missing.

9. In the experiment, the repeat times seems to be small (only 10 runs are conducted). In some settings, the standard deviation of the results is 0. How does that come?


**Limitations:**

The authors mention about the limitation that the method is specifically tailored to datasets consisting in dyadic data for biclustering and can not be applied on other data types such as images directly.

**Strengths And Weaknesses:**

originality
Strength
This paper leverages the low-rank optimal transport to solve the biclustering problem for the dyadic data.

quality
The paper is generally well written in grammar. However, the technical discussion lacks clarity in some respects. Such as Page 1 line 27, the “summary matrix” is mentioned yet without further explanation.

clarity
Weaknesses
(1)	It is unclear what’s the major advantage makes the proposed methods perform better than the others in the experiments.
(2)	A lot of details of the experiment seem to be missing. Such as, how are the parameters r and c selected. There are lots of existing works introduced in section 3 seems can solve the discussed problem also and what they are not compared.
significance
This paper leverages the optimal transport to solve the biclustering problem for the dyadic data. The computational efficiency and estimation accuracy are achieved through the low-rank assumption of the solution matrix.

---

> ### Author Response · Authors · 2022-08-02
> **Strengths of BCOT for biclustering**
>
> Thank you for your thorough review and for supporting our proposal. We appreciate the comments and suggestions that we are willing to address in the revised version. A first revised version is already available.
>
>
> # Addressing the perceived weaknesses
> 1. BCOT, COOT and CCOT tackle the biclustering problem in very different manners.
>
> With the COOT and CCOT variants, a co-clustering is proposed at the convergence ; co-clustering is then a consequence and not a main goal. However, BCOT aims at co-clustering and integrates this objective from the beginning and not to deduce it at the end of any process.
> Unlike the COOT and CCOT variants, with BCOT we keep the original data as input.
> We propose a general formulation for the block seriation which reveals a link with low-rank optimal transport formulated as a minimization linear program easier to solve. The approach has the added possibility of choosing the distribution of elements over the row and column clusters (through setting the proportional size of clusters using r and c).
> Further, unlike COOT and CCOT, BCOT implicitly performs dimensionality reduction dealing thus effectively with sparse and noisy high-dimensional data. At each iteration row clusters are updated based on low dimensional representation spanned by the  column clusters $L(B)W$ and vice-versa by the row clusters $L(B)^T Z$ showing, thereby, the mutual reinforcement  between row and column clusterings.
> Faced with real data, in the literature when we deal with co-clustering we unfortunately limit ourselves to an evaluation of a one-side clustering (document clustering for instance), with BCOT we show the consistency of biclusters by dealing with term clustering.
>
> This explains the interest of our approach.
>
> 2. $r$ and $c$  should be set to the desired proportion of the clusters. For example r=(0.3, 0.7) means that the first cluster should contain around 30% of the row elements while the second 70%. The same reasoning applies for $c$ wrt the column elements. As mentioned in the paper, when no such information is available, setting them to the uniform distribution is the most reasonable choice.
>
> 3. Thank you for this remark, the established connections (section 3) lead us towards two interesting lines of comparison, the first by a simple comparison with original algorithms optimizing the criteria (10, 11, 12); we intend to add the results showing that BCOT outperforms them. The second being based on new versions of BCOT, by setting the function $L(B)$ accordingly, should be interesting to investigate.
>
> # Answers to questions
>
> 1. Please see 1. in # Addressing the perceived weaknesses
>
> 2,3,4. The three constraints (binarity, assignment, impossible triads) are implicitly included in (5) since $Z$ and $W$ are classification matrices. Matrix $C=ZW^\top$ respects all three constraints as well as the rank constraint.
>
> 5. We defined the discrepancy as a sort of reciprocal to connectivity meaning that the largest entries in the adjacency matrix should become the smallest entries.
>
> 6. The dimensions of the distributions $\mathbf{r}$ and $\mathbf{c}$ are the same, they are both equal to $k$, the desired number of biclusters, it is their entries that are not necessarily the same.
>
> 7. We will revise this in the final version.
>
> 8. They are different objective functions, BCOT$_\lambda$ (8) is to the fuzzy block seriation problem (9) what BCOT (6) is to the block seriation problem (5). We introduced (9) due the fact that there was no prior concept of a fuzzy block seriation in literature (to the best of our knowledge).
>
> 9. This is due to the fact that the authors in [19] omitted the computational complexity associated with computing two pairwise distance matrices, one over the rows and one over the columns (as seen in **algorithm 1 and 2** in [19]).
>
> 10. If we know the distribution of the cluster sizes over the rows and columns, we can set $\mathbf{r}$ and $\mathbf{c}$, otherwise, it is better to use a uniform distribution. $k$ is the same as the number of biclusters which is required as a hyper-parameter.
>
> 11. We have increased the number of runs and the results are similar. The presence of sd=0 means that over several runs the algorithm leads to the same co-clustering.

---

> > ### Comment · Reviewer_Piwj · 2022-08-09
> > **I will maintain my review score**
> >
> > I thank the authors for their comments. Please revise the mentioned part in the manuscript and probably add some more details about the computational complexity (answer 9) in the manuscript. The sd =0 still look suspicious and need more clarifications.

---

### Official Review · Reviewer_3Scn · 2022-07-23

**Rating:** 4
**Confidence:** 3
**Soundness:** 2 fair
**Presentation:** 1 poor
**Contribution:** 2 fair

**Summary:**

In the given paper, the authors propose a novel biclustering algorithm for dyadic data based on optimal transport (OT) and the block seriation problem. Their main contribution can be summarized as the formulation of the optimization problem BCOT, an indefinite bilinear program, whose solution can be used to obtain biclusters, i.e., a simultaneous clustering of rows and columns. In special cases, the solution of BCOT allows for the computation of an approximation of the optimal transport map with bounded rank for the discrete Kantorovich OT formulation. In addition to BCOT, the authors also introduce the fuzzy variant BCOT$_{\lambda}$, which is an adaptation of BCOT with entropic regularization, and connect BCOT to existing biclustering algorithms. To empirically evaluate the proposed method, the authors perform experiments with six document-term datasets and compare their results with four other OT based biclustering approaches.

**Questions:**

1. Could you please elaborate on the row exemplar and column exemplar distributions? How do they relate to the row and column weights? In [26] these are distributions from the second data matrix, but what are their purpose in your work?

2. Could you give more details on the approximation of the optimal transport map (lines 136-139)? Also, what do we know about the rank of $Z$ and $W$?

3. Table 1: How did you compute the time complexities for CCOT, CCOT-GW [19], COOT and COOT$_{\lambda}$ [26]? Why do they differ from the time complexities reported in [19] and [26], respectively? What restricted class of cost functions is considered?

4. Empirical evaluation

    4.1. Have you considered using simulated data to have ground truth biclusters, which you can use to assess your method? This is quite common for the evaluation of novel biclustering methods, and is also done in [19] and [26]. If it is due to the page limit, you could also consider including this in the appendix or supplementary material.

    4.2. What is the reasoning for the choice of datasets? Why mainly document-term matrices and not e.g., gene expression data?

    4.3. It would be beneficial to include the majority of baselines also used in [19] and [26]: ITCC, Double K-Means, Orthogonal Nonnegative Matrix, Tri-Factorizations (ONTMF), the Gaussian Latent Block Models (GLBM) and Residual Bayesian Co-Clustering (RBC).

    4.4. Why did you choose to compare your method, which is specifically designed to work well for dyadic data, with methods which are not? Have you considered including common biclustering methods for dyadic data as baselines?

    4.5. Why did you look at document and term clusters separately, and not evaluate your obtained biclusters using e.g., the CCE (see point 4.4)? If you lack ground truth data, please consider point 4.1.  It might be beneficial to compare your method with CCOT, CCOT-GW [19], COOT, COOT$_{\lambda}$ [26] with respect to the CCE as well.

5. Applications

    5.1. Reference [30]: What applications of biclustering do the authors discuss in this book? Could you please give some pointer to specific pages?

    5.2. As the main novelty in this contribution seems to be the focus on dyadic datasets, could you give an outlook on other areas of application (in addition to document-term matrices) and elaborate on the significance of your contributions?

**Limitations:**

The authors addressed the limitations of their work pertaining to the type of data suitable for their proposed method.

**Strengths And Weaknesses:**

**Strengths**

_Originality_: The concept to combine the block seriation problem and optimal transport to obtain block diagonal biclusters for dyadic data is a novel idea.

_Quality_: In the presented work, I did not see any major technical issues and the authors provide proofs for the propositions in the appendix.

_Clarity_: The overall structuring of the paper is good, especially section 3 (_Connections to Existing Work_), as it helps to contextualize the proposed method with respect to related work. Concerning reproducibility, the authors publicly released their code. The application of a statistical test to quantify the significance of the experimental results is a nice addition.

_Significance_: The presented results show that the proposed method is superior to existing OT based biclustering methods with respect to document-term clustering.

**Weaknesses**

_Originality_: Although the presented method is novel, it heavily relies on previous work (i.e., [19, 21, 26]) and provides comparably little conceptual originality. The contribution's main novelty seems to be that it works better for dyadic data compared to other OT based biclustering methods. As it is conceptually not that convincing, I would consider it more of an empirical paper, and for that it lacks sufficient experimental evaluation with SOTA biclustering methods for dyadic data.

_Quality_: While there are no obvious major technical errors, there are some vague statements and (suspected) smaller errors in some of the formulas. Moreover, instead of using bicluster-specific metrics, such as e.g., the co-clustering error (CCE), the authors only assessed the resulting clusters using standard clustering metrics. Overall, I am also missing a clear motivation, other than the fact that optimal transport is a trending topic in the machine learning community.

_Clarity_: One of my main concerns is that the paper (in the given version) is not sufficiently self-contained. It was difficult to understand the presented content without consistently referring to [19] and [26] and further referenced literature. Also, a clear train of thought is missing throughout the paper (e.g., in Section 2.2 we switch from inducing a biclustering based on the optimal transport block seriation problem to an approximation of the Kantorovich OT formulation without much transition), which is especially evident in the introduction. I would suggest starting with a more general motivation before going into detail about bipartite graphs as well as applications. While contextualizing the work in section 3 is helpful, this section is, again, not fully self-contained and comprehensible, and related work needs to be frequented. Concerning language and grammar, there is room for improvement. Language is oftentimes repetitive, somewhat colloquial, and not always precise.

_Significance_: While the BCOT performs much better to identify document and term clusters in comparison to existing OT based biclustering methods, the overall significance of these results is not (yet) clear to me. I have multiple questions/concerns about the methodology of the empirical evaluation, which includes, among others, the choice of datasets, baselines, and metrics (please see **Questions 4.x** for more details).

Please see below for a more detailed feedback.

**Minor remarks & suggestions**

* References:
    - Please consider citing the peer-reviewed version instead of the arXiv version in the references (e.g. reference [24])
    - Lines 30-31: Please consider adding references that support your statement ("Optimal Transport (OT) took the machine learning community by storm and was used in the resolution of various data mining problems and biclustering was not an exception")
    - Lines 42-43: Please consider adding references for the previous biclustering approaches, which your method generalizes
    - Lines 61-63: Please consider adding a reference for the the Kantorovich formulation of OT
    - Line 225: Please consider adding a reference for the Davies-Bouldin index
    - Line 233: Please consider adding references for the employed clustering metrics

* Structure:
    - Please consider restructuring your introduction, starting with a more general motivation and introduction to biclustering and OT. Also, pointers to sections would be much appreciated (e.g., "In Section 2, we propose our method...").
    - Lines 25-29: Some more high-level classification of biclustering algorithms might be helpful for the reader (e.g., there exist probabilistic approaches, etc.) in addition to enumerating applications.

* Language/Grammar:
    - Please consider refraining from very long sentences (e.g., lines 32-35, 140-143)
    - Lines 103-104 is not a proper sentence
    - Line 145: mentionned -> mentioned
    - Line 170: "proposed to fuzzy variant" -> "proposed a fuzzy variant"
    - Line 171: traditionanal -> traditional
    - Line 191: in -> is
    - Line 194: "algorithm available" -> "algorithm is available"
    - Line 198: "in a way that make" -> "in a way that makes"
    - Line 223: $\lambda$ is not in subscript
    - Line 282: "certain types dyadic data" -> "certain types of dyadic data"

* Examples for vague language:
    - Lines 43-44: "We propose two efficient methods for solving this problem [...]" -> What problem?
    - Line 98: "Now let its [...]" -> What is _it_?
    - Line 107: "[...] we are interested in inducing a couple of _almost-hard clustering_ [...]" -> What does _a couple_ refer to?
    - Lines 117-118: "[...] this should not significantly change the structure of the solution [...]"
    - Lines 130-131: "[...] row exemplars (or representatives or centroids)" -> Please consider using one of these terms consistently
    - Line 134: "Biclustering is the main purpose of the approach we proposed [...]" -> Could you be more precise what is meant by _the approach we proposed_
    - Line 200: "[...] the computation of [...] is quite efficient [...]"
    - Line 209: "[...] our model should be faster in most cases [...]"
    - Line 286: "[...] the proposed approach does a good job of finding clusters [...]"

* (Suspected) minor errors:
    - Line 70: Is $\mathbf{K}=-\texttt{diag}(\mathbf{a})\exp(\mathbf{M}/\lambda)\texttt{diag}(\mathbf{b})$ correct, cf. [4], Lemma 2? Should it be a negative exponential, i.e., $\exp(-\mathbf{M}/\lambda)$?
     - Lines 78-79: "The block seriation problem is an integer programming problem and is consequently NP-hard" -> Please consider adding a reference which contains a proof for NP-hardness for the block seriation problem
    - Line 99: Is $\mathbf{Z}$ a typo?
    - Line 158: $\sum_{h=1}^r p(\mathbf{b}_i, \mathbf{b}'_j \in h)$ -> Should it be "$=$" instead of "$\in$"?

---

> ### Author Response · Authors · 2022-08-02
> **BCOT versus COOT and CCOT  to tackle biclustering**
>
> Thank you for your review and appreciate the comments and suggestions that we are willing to address in the revised version
>
> #Originality
>
> We respectfully disagree with the assessment that our work is heavily based on [19,21,26].
> - [19] identify biclusters by detecting jumps in the scaling vectors \alpha and \beta in
> the solution of the entropic regularized OT in CCOT and the entropic Gromov-Wasserstein in CCOT-GW
> - [26] rely on Information-Theoretic CoClustering whereby a summary matrix of size $k\times g$ which is as close as possible to the original data matrix wrt loss in mutual information is learned to perform biclustering. The same idea is applied in COOT except that they minimize the COOT metric instead
> - [21] is nothing more than a problem statement for biclustering as a maximization of integer programming
>
> While in COOT and CCOT, a co-clustering is proposed at the convergence of the criterion (co-clustering is a consequence and not the main goal), BCOT aims at co-clustering and integrates this objective from the beginning. This explains the interest of our approach. Thereby, we propose a general formulation for the block seriation which reveals a link with low-rank optimal transport formulated as a minimization linear program easier to solve. The approach has the added possibility of choosing the distribution of elements over the row and column clusters (through setting the proportional size of clusters using r and c). Further, unlike COOT and CCOT, BCOT implicitly performs dimensionality reduction dealing thus effectively with sparse and noisy high-dimensional data. At each iteration row clusters are updated based on low dimensional representation spanned by the  column clusters  $L(B)W$ and vice-versa by the row clusters $L(B)^T Z$ showing, thereby, the mutual reinforcement  between row and column clusterings
>
> #Quality
> CCE is used for simulated data because of the availability of labels of row and column clusters.  In our tests, only the row/document labels are present and so we propose to use accuracy for row clusters and PMI-based score to evaluate the column clusters’ coherence.
> #Clarity
> The transition between the presentation of biclustering and the OT parts seems sudden
> because there is no inherent connection between the two and the connection becomes clear once we delve into the details of the proposed BCOT problem
>
> #Answers
>
> Points 1-9 are summarized in 1-3
> 1. The row and column exemplar distributions r and c can be seen as the distribution of the row and column clusters respectively. There is no inherent relation between them and row and column weights just like for the source and target distributions in optimal transport.
> 2. A detailed discussion of the approximation is out of the scope of this paper which focussed mainly on the biclustering part; Z, W are not necessarily full-rank
> 3. For the COOT variants the complexities are the same as the one reported in [26], we only adapted them to biclustering (d’=k,n’=k). Furthermore, the complexity they reported is only for the computation of $L(X,X^T)$, we thus added the additional complexity of the iterative part of their BCD algorithm. For CCOT, the authors omitted the complexity of initial computation of two pairwise distance matrices over the rows and the columns
>
> 4. 4.1 The simulated data according to a Gaussian LBM [19,26] are small in size with few row and column classes ; there is no information on the parameters used to generate them. As we are interested in dyadic data (sparse/not), not to mention that all values are positive, it is certainly not the GLBM to be recommended for this task
>
> 4.2. We chose document-term matrices because 1) It is mainly on this type of matrices (high dimensional and sparse) that co-clustering has convinced the most the ML community on its interest compared to clustering 2) the evaluation of new approaches is more straightforward due to the presence of document labels and the possibility of using semantic coherence that we propose 3) Our expertise in co-clustering showed that processing gene expression data is not the same as processing document-term data which are sparse; the underlying models are not the same
>
> 4.3. We add two baselines, ITCC and ONMTF; most suited for document-term matrix and competitive unlike the others. BCOT outperforms ITCC and ONMTF. Note that BCOT generalizes multiple biclustering models and so should be as effective as these models when using the corresponding L function (section 3). In [19,26] RBC, GLBM and DKM make the Gaussian assumption, and are therefore not appropriate for document-term. Further, the use of ITCC and ONMTF in [19,26], requiring that all values are non-negative, is inappropriate
>
> 4.4. Please see 4.3
>
> 4.5.  Please, see the #Quality section
>
> 5.1. We cite [30] for possible applications of biclustering rather than clustering on dyadic data
>
> 5.2 Dyadic data appear in survey research, marketing, business intelligence, information retrieval, and recommender systems

---

> > ### Comment · Reviewer_3Scn · 2022-08-08
> > **Remaining questions**
> >
> > Dear Authors,
> >
> > thank you for your comments and clarifications!
> >
> > Some concerns and/or questions regarding the empirical methodology remain:
> >
> > * **Ground-truth data** (partially addressed in **4.1**): What is your reasoning to not simulate synthetic dyadic data suitable for BCOT and BCOT$\_{\lambda}$? In this way, you could compare BCOT and BCOT$\_{\lambda}$ to the other methods with respect to retrieving biclusters, which could strengthen your empirical evaluation.
> > * **Choice of datasets** (addressed in **4.2**): As you mention in the introduction, the analysis of gene expression data is one of the main application areas of biclustering; many biclustering methods are developed in this context, and, arguably, the field of computational biology is also interesting for the broader ML community. Could you clarify what you mean with "Our expertise in co-clustering showed that processing gene expression data is not the same as processing document-term data which are sparse; the underlying models are not the same"?
> > * **Baselines**: Thank you for adding the additional baselines for document clustering. It is interesting to see that ITCC is fairly competitive with BCOT and BCOT$\_{\lambda}$, how does it compare in runtime? Also, in your answer to Reviewer _c4gm_, you mention that you optimized the hyperparameters of your method according to "rules of thumb and internal unsupervised metrics such as the Davies-Bouldin index", but for the baseline methods, you used "default values prescribed by the authors of each baseline". Have you tried performing hyperparameter tuning for the baselines? It would be worth investigating whether ITCC is able to outperform BCOT and/or BCOT$\_{\lambda}$ with some hyperparameter optimization.

---

> > > ### Author Response · Authors · 2022-08-09
> > > **Answers to the remaining questions**
> > >
> > > We thank you for your response and the interest you show in bettering our paper!
> > > # Ground-truth data
> > > We have performed additional experiments on the following synthetic datasets:
> > >
> > > ||rows|cols|biclusters|bicluster sizes|Sparse|Structure|
> > > |-|:-:|:-:|:-:|:-:|:-:|:-:|
> > > |A|500|500|10|equal|Yes|Block diagonal|
> > > |B|800|1000|6|unequal|No|Block diagonal|
> > > |C|800|800|7|equal|No|Checkerboard|
> > > |D|2000|1200|4|unequal|No|Checkerboard|
> > >
> > > We have generated A in a way as to make it similar to doc-term matrices (sparsity). B, C and D are made to be similar to gene-expression data (matrices containing biclusters with somewhat constant values). You can see the structure of these models in the following figure: https://imgur.com/a/VW21nLI
> > >
> > > The results are averaged over 10 runs. The metric used is **(1-CCE)$\times$100**
> > > | | A | B | C | D |
> > > |-|:-:|:-:|:-:|:-:|
> > > ITCC|80.1±6.1|93.8±6.7|91.1±4.6|97.1±1.7|
> > > CCOT|54.4±3.5 | 70.0±.0 | 29.7±.4| 55.7±1.8 |
> > > CCOT-GW|99.1±.0|83.5±.0|83.4±.0|75.3±.0|
> > > COOT|99.8±.0|78.8±2.0|99.8±.0|93.7±1.2|92.2±1.1|
> > > COOTλ|39.9±2.4|84.9±4.6|28.2±.0|60.7±.0|39.5±1.9|
> > > BCOT | 99.8±.0|80.4±2.2|99.6±.1|91.3±.7|
> > > BCOTλ |**100±.0**|99.1±.4|**100±.0**|**100±.0**|
> > > BCOT (ground truth **r** and **c**)|same|99.9±.0|same|95.5±2.3|
> > > BCOTλ (ground truth **r** and **c**) |same|**100±.0**|same|99.2±.9|
> > >
> > > Our models have the best results on all four datasets (tie with COOT on A). We thank you as these tests additionally allow us to show the utility of the the row cluster distribution **r** and column cluster distribution **c**. The use of these ground truth distributions resulted in an increase of 19.5 and 4.2 points for BCOT on C and D; and an increase of .3 and decrease of .8 for  BCOTλ.
> > >
> > > #  Choice of datasets
> > > In bioinformatics, the homogeneity of a bicluster or several biclusters depends on the sought after model and has to respect some constraints. For instance, this is illustrated in figure 1 of Madeira, S.C., & Oliveira, A.L. (2004). Biclustering algorithms for biological data analysis: a survey. TCBB, 1(1), 24-45. The discussed models and algorithms are devoted to the biclustering task. This is not the same problem searched in document/term clustering where we aim to reveal groups of documents characterized by groups of terms. Thereby, we are concerned with the co-clustering of sparse high dimensional data. When dealing with such data, seeking homogeneous blocks may not always be enough to produce useful, ready-to-use results. In fact due to data sparsity, several co-clusters may, for example, be primarily composed of zeros. Such co-clusters, while homogeneous, are not relevant and must be filtered out in the post-processing phase. In other words, it is for the user to select the most useful co-clusters so as to determine which document clusters should go with which term clusters, a task which is, however, not straightforward even with a reasonable number of document and term clusters. Approaches that take into account the sparsity characteristic exhibited by text data are, therefore, needed  if co-clustering approaches are to be usable in realistic scenarios. This is the aim of our proposal for document and term clustering; it has the advantage of directly producing the most meaningful co-clusters.
> > >
> > > **We have also added two gene-expression data benchmarks, the CuMiDa breast cancer (Breast_GSE57297) and Leukemia datasets (Leukemia_GSE9476). BCOTλ has the best performance on both of them:**
> > >
> > > |||Breast Cancer|||Leukemia||
> > > |-|:-:|:-:|:-:|:-:|:-:|:-:|
> > > ||ACC|NMI|ARI|ACC|NMI|ARI|
> > > ITCC|68.5±11.9|24.2±16.8|16.5±15.4|64.7±7.4|61.9±7.0|39.8±7.6|
> > > CCOT||OOM||40.6±.0|.0±.0|.0±.0|
> > > CCOT-GW||OOM|||OOM||
> > > COOT|63.1±5.2|5.4±8.7|-1.2±2.9|36.2±2.7|14.0±3.6|5.4±3.2|
> > > COOTλ|61.5±.0|5.4±.0|2.2±.0|32.5±3.3|8.7±2.7|-.5±2.1|35.9±1.7|9.8±5.5|1.4±1.8|
> > > BCOT|76.9±.0|37.2±.0|26.7±.0|71.2±5.4|59.6±6.9|39.9±6.3|
> > > BCOTλ|**84.6±.0**|**48.3±.0**|**46.0±.0**|**80.9±3.8**|**70.9±4.1**|**55.3±3.3**|
> > > # Baselines
> > > Here is a table containing the training times of BCOT, BCOTλ and ITCC. One of our two models has the fastest training times on the four datasets (the shortest time is highlighted in bold). All models had the same number of iterations and results are averages of 10 runs.
> > > Method|ACM|DBLP|Pubmed|Wiki|
> > > |-|:-:|:-:|:-:|:-:|
> > > ITCC|1.53±.46|.88±.23|4.42±1.07|5.66±.98|
> > > BCOT|.93±.36|**.74±.25**|7.97±.72|6.01±.69|
> > > BCOTλ|**.64±.19**|4.56±.45|**2.98±.31**|**5.6±.74**|
> > >
> > > ITCC requires only the number of row and column clusters as hyperparameters. Unlike BCOT; with COOT, CCOT and ITCC the number of row and column clusters is not necessarily the same. However, in the case of sparse data for example, by seeking to reveal a block diagonal structure (biclustering with the same number of row and column clusters), BCOT filters out homogeneous but noisy blocks making the results easier to analyze and interpret. Note that ITCC deals with nonnegative matrices only.
> > >
> > > **These experiments will be added to the supplementary material. We hope that we have addressed the reviewer's concerns!**

---

### Meta-Review · Area_Chair_LHud · 2022-08-23

**Recommendation:** Accept
**Confidence:** Certain

**Metareview:**

The reviewers discussed strengths and weaknesses of the paper. One potential issue (to which the author's answer was rather unhelpful) was resolved by a reviewer running the experiments with higher precision output. Reviewers were mostly convinced by the strong empirical improvements.


**Award:**

No

---

### Decision · Program_Chairs · 2022-09-14

Accept